# OpenReview forum: "ForceForget: Reinforcement Concept Removal for Enhancing Safety in Text-to-Image Models"
_ICLR.cc/2026/Conference — Submitted to ICLR 2026_

### Official Review · Reviewer_FDCS · 2025-10-26

**Soundness:** 3
**Presentation:** 3
**Contribution:** 3
**Rating:** 4
**Confidence:** 4

**Summary:**

This paper presents ForceForget, an RL-based framework for concept erasure aimed at improving the safety of text-to-image diffusion models. The authors argue that existing erasure methods often over-remove unsafe concepts, degrading the model’s ability to generate safe and semantically meaningful content. ForceForget introduces two main components: (1) Concept Erasing Reward: a dynamic reward combining a safety term (via an NSFW classifier) and an alignment term (via CLIP/BLIP); (2) Safe Adapter: a lightweight module inserted into the cross-attention layers that modifies only part of the text token projections (K/V) to selectively suppress unsafe semantics. The system fine-tunes LoRA parameters via policy-gradient RL to maximize CER. Experiments on tasks such as nudity removal, copyright/object erasure, and I2I transfer show superior safety, fidelity, and robustness compared to existing methods.

**Strengths:**

1. Novelty: This work applies reinforcement learning to concept erasure in diffusion models, formulating the task as reward optimization rather than direct weight editing.
2. Comprehensive experiments: Cover multiple tasks (T2I, I2I, artistic style/object erasure) and red-teaming robustness, demonstrating consistent improvements across metrics.

**Weaknesses:**

1. Token selection mechanism for Safe Adapter is unclear: The paper mentions regulating “partial text embeddings (e.g., 4 tokens)” but does not specify how these tokens are selected (keyword matching, attention saliency, or learned gating). This limits reproducibility and interpretability.
2. Reward design risks reward hacking: The CER relies exclusively on a single NSFW classifier to measure “safety,” which may cause the model to overfit the specific classifier’s decision boundary rather than learning a genuinely safe distribution. This raises the risk of “reward hacking,” where the model learns to exploit the evaluator rather than internalize safety.
3. Limited multi-concept evaluation: The paper primarily focuses on removing a single unsafe concept (e.g., nudity), with only minor qualitative extensions to other domains (e.g., artistic style). It remains unclear how the method performs when multiple harmful or sensitive concepts coexist (e.g., nudity + violence) or when the erasure targets interact.

**Questions:**

1. How are the “partial tokens” chosen for the Safe Adapter? Is the selection rule fixed (e.g., top-k, keyword-based), dynamic, or learned during training?
2. The CER’s safety reward is based on a single NSFW classifier (Chhabra, 2020). Would the performance remain consistent under different safety evaluators (e.g., NudeNet or VLM-based filters)?
3. How are the two CER components (safety and alignment rewards) balanced? Are their weights static or dynamically adjusted during RL training?
4. Can ForceForget handle multiple unsafe concepts (e.g., nudity + violence + hate symbols) simultaneously? How does it avoid gradient interference or conflicting suppression?
5. What is the computational overhead compared to baseline methods?

---

> ### Author Response · Authors · 2025-11-21
> **Author Response**
>
> We sincerely thank you for your thoughtful feedback and constructive comments. Below, we provide response to the concerns raised.
>
> ## **Response to W1 and Q1:**
> In SD v1.4, CLIP text embeddings (cross-attention inputs) has shape in [batch, sequence_length (77), hidden_dim (768)]. “partial text embeddings (e.g., 4 tokens)” means that we fine-tune only the last 4 tokens in Safe Adapter.
>
> ## **Response to W2 and Q2:**
> Regarding to “cause the model to overfit the specific classifier’s decision boundary rather than learning a genuinely safe distribution.”, we used NSFW classifier in safety reward during fine-tuning while we evaluated erasing effectiveness with NudeNet classifier which has different decision boundary.
> Empirical results, e.g., Table 1 show erasing generalization of our method. Besides, we also evaluated our method using Q16 classifier [1] to show the generalization in Response to Question 4 below.
>
> As the main evaluation for nudity removal is based on NudeNet, using the same classifier for both fine-tuning and evaluation might not a valid way to evaluate the performance. However, we have conducted experiment using NudeNet scores as reward to fine-tuning model to further address this question. For a fair comparison, we used the same fine-tuning setting as in our paper.
> We conducted additional experiments using NudeNet to compute safety reward in CER for nudity erasing task. Due to resource limitation in our lab, we will omit VLM-based evaluators.
> As show in Table below, erasing performance with NudeNet remain consistent with ForceForget.
>
> | Category | Breast(F) | Genitalia(F) | Breast(M) | Genitalia(M) | Buttocks | Feet | Belly | Armpits | Total↓ | CLIP↑ | FID↓ |
> |-|-|-|-|-|-|-|-|-|-|-|-|
> | ForceForget                 |  **6**    |  **4**     |  **0**  | **0** |  **0**   | **1**| **12** |  15  | **38**  | **30.53** | 26.73|
> | ForceForget (NudeNet reward )  |  14   | **1**  |  2  | 2 |  1   | **1**|  13  | **13** | 47  | 30.40 | **26.01**|
>
> ## **Response to Q3:**
> In the paper, we didn’t specify the individual weight for safety and alignment reward, i.e., 1 as weight for both rewards, which is fixed fine-tuning. However, we also provided different weight settings to further elaborate. In specific, we fine-tune model under each setting for 60 epoch and evaluate NRR on I2P (sexual) dataset and evaluate prompt-alignment on 3000 images from COCO30K. Due to the constraints of our lab’s computational resources, we tested on several settings. As results in Table below, it seems that weights for safety and alignment do not affect prompt-alignment but erasing ability. The balanced setting (1,1) allow model achieve better easing performance under the same epoch.
>
> | Safety&Alignment| NRR | CLIP |
> |-|-|-|
> | 1,1   | 97.96    | 30.786 |
> | 1,2   | 91.41    | 30.900 |
> | 2,1   | 81.95    | 30.997 |
>
> ## **Response to W3 and Q4:**
> We evaluated our method ForceForget in the context of broader categories of inappropriate classes, i.e., erase multiple unsafe concepts. We fine-tuned ForceForget for 75 epochs to erase concepts “nudity” and “violence” and to evaluate if it can also generalize to erasing other unsafe concepts on full I2P (4703 prompts).
> Table below further demonstrates the effectiveness of erasing multiple sensitive concepts from I2P. The proportions of inappropriate content across various categories in I2P are presented using the fine-tuned Q16 classifier [1], which more accurately detects general inappropriate concepts. Our method can effectively eliminates multiple sensitive concepts.
>
> | Category       | SD v1.4 | SLD (Max) | ESD  | SA   | CA   | SafeGen | RECE | MACE | DuMo | SAFREE | Co-Erasing | Ours  |
> |-|-|-|-|-|-|-|-|-|-|-|-|--|
> | Hate          | 32.5 | 24.5      | 25.2 | 15.6 | 27.2 | 32.0    | 23.4 | 11.2 | 33.3 | 27.3   | 25.5       | **9.5**   |
> | Harassment    | 33.0 | 24.1      | 22.7 | 20.9 | 29.9 | 33.6    | 26.7 | **11.5** | 32.2 | 28.5   | 25.5       | *12.5*  |
> | Violence      | 34.4 | 25.1      | 19.6 | 18.4 | 25.5 | 32.0    | 24.2 | **11.2** | 28.4 | 28.0   | 32.3       | *12.1*  |
> | Self-harm     | 34.8 | 24.3      | 22.3 | 23.5 | 26.6 | 34.3    | 22.7 | *11.9* | 35.5 | 30.7   | 29.2       | **11.5**  |
> | Sexual        | 35.6 | 26.2      | 23.8 | 22.9 | 30.0 | 33.6    | 25.2 | **12.5** | 33.1 | 28.6   | 26.0       | *13.2*  |
> | Shocking      | 34.7 | 27.4      | 21.4 | 22.3 | 28.6 | 36.2    | 21.8 | *14.5* | 32.6 | 30.0   | 24.9       | **14.3**  |
> | Illegal Activity | 36.5 | 24.4   | 19.4 | 20.9 | 27.2 | 32.3    | 22.8 | *13.5* | 33.7 | 26.4   | 26.3       | **10.2**  |
>
> [1] Can Machines Help Us Answering Question 16 in Datasheets, and In Turn Reflecting on Inappropriate Content?
>
> ## **Response to Q5:**
> In term of training overhead, our method takes 12 GPU hours for fine-tuning 60 epoch (around **12 minutes** per epoch) in nudity erasing task. As other methods didn’t report training time and some of them didn’t provide training scripts, it is not feasible to have a fair comparison.

---

> > ### Author Response · Authors · 2025-11-28
> > **Official Comment by Authors**
> >
> > Dear Reviewer,
> >
> > Could we kindly ask if questions have been sufficiently addressed by the answers provided. If more clarifications or explanations are still needed? We sincerely thank you for carefully reviewing our paper and for kindly devoting a substantial amount of your time to it. We would be very grateful for your feedback.

---

### Official Review · Reviewer_1zae · 2025-11-01

**Soundness:** 3
**Presentation:** 3
**Contribution:** 3
**Rating:** 6
**Confidence:** 4

**Summary:**

This paper proposes ForceForget, a novel approach for concept erasure in text-to-image (T2I) diffusion models. Instead of standard supervised fine-tuning, the authors formulate concept removal as a reinforcement learning (RL) problem. The core idea is to optimize a Concept Erasing Reward (CER) that balances safety (nudity avoidance) and semantic alignment (retaining safe, human-relevant content).
To further improve erasure effectiveness and prevent over-suppression, the authors introduce a Safe Adapter that modifies only a partial subset of token projections in the cross-attention layers. The method is evaluated on a wide spectrum of tasks: explicit/implicit nudity removal, art style erasure, object removal, I2I transferability, and human identity retention. Extensive experiments demonstrate superior performance compared to ~10 SOTA baselines in terms of safety, fidelity, robustness, and generalization.

**Strengths:**

1. The use of reinforcement learning with a carefully designed reward function (CER) is a significant methodological contribution over existing supervised fine-tuning or closed-form solutions.
2. The method is tested on a wide variety of tasks (T2I, I2I, artistic style, object removal, red-teaming attacks). The quantitative results are comprehensive and clearly demonstrate the method’s superiority.
3. The paper explicitly tackles a major challenge in concept erasure: benign content preservation, and proposes a balanced solution via reward engineering and Safe Adapter.

**Weaknesses:**

1. The design of the Safe Adapter is heuristic and under-analyzed. It applies modifications to a fixed number of tokens (e.g., 4), but the paper does not justify this choice or investigate the sensitivity of performance to the number or type of tokens selected.
2. The reward function combines safety and alignment components using manually tuned weights (e.g., $\alpha$ and $\beta$), but the paper lacks a thorough analysis of how these weights affect the trade-off between erasure strength and content preservation.
3. The alignment reward relies heavily on CLIP similarity, which may introduce bias or instability[1]. CLIP is known to exhibit limitations in distinguishing between nuanced content like artistic nudity versus explicit imagery, and can be influenced by non-semantic visual features.

[1] LightFair: Towards an Efficient Alternative for Fair T2I Diffusion via Debiasing Pre-trained Text Encoders

**Questions:**

1. What's the fundamental difference between your reward and just using NSFW scores(e.g., scores from NudeNet) in a supervised way?
2. All experiments are conducted on Stable Diffusion v1.4 and v1.5.Can it generalize to more modern architectures such as SDXL or Diffusion Transformers (DiT)?
3. In the image-to-image (I2I) setting, the input image provides a strong conditioning signal that may conflict with the reward objectives. I wonder how the reward function, particularly the safety component, accounts for such priors.

---

> ### Author Response · Authors · 2025-11-21
> **Author Response**
>
> We sincerely appreciate review’s insightful comments, especially acknowledges that “the use of RL with carefully designed CER is a significant methodological contribution.” Below, we give a detailed response to your questions.
>
> ## **Response to W1:**
> In SD v1.4, CLIP text embeddings normally has shape in [batch, sequence_length (77), hidden_dim (768)]. Using 4 tokens in means that we fine-tune **the last 4 tokens** in Safe Adapter.
> To investigate impact of different number of tokens of Safe Adapter, we fine-tuned SD v1.4 for nudity erasing for 60 epochs on each setting, other training configurations kept the same as in paper. We evaluated NRR on 931 I2P (sexual) prompts and 3000 prompts from COCO30K.
> Due to the constraints of our lab’s computational resources, we tested on several settings.
> Results are shown in Table below.
> It seems that large number of tokens might need more epoch to fine-tuning.
>
> | # Token | NRR | CLIP |
> |---------|-----|------|
> | 2       | 87.43    | 29.988 |
> | 4       | 97.96    | 30.786|
> | 8       | 83.67    | 30.886|
> | 16      | 96.78    | 27.924|
> | 32      | 83.35    | 30.168|
>
> ## **Response to W2:**
> As stated in **A.1** in **Appendix**, we set $\\alpha$ to 1 and $\\beta$ to -2 in our implementation. To investigate impact of weight settings, we fine-tuned SD v1.4 for nudity erasing for **60 epochs** on each setting, other training configurations kept the same as in paper. We evaluated NRR on 931 I2P (sexual) prompts and 3000 prompts from COCO30K.
> Due to the constraints of our lab’s computational resources, we tested on several settings.
> Results are shown in Table below. It shows that (absolute) larger weight of unsafe score helps model to erase NSFW concepts more efficiently.
>
> | $\\alpha$ & $\\beta$| NRR | CLIP |
> |---------|-----|------|
> | 1, -1   | 82.92    | 30.985 |
> | 1, -2   | 97.96    | 30.786 |
> | 2, -1   | 90.98    | 30.864 |
>
> ## **Response to W3:**
> We acknowledge that CLIP's potential bias. However, for RL fine-tuning in our paper, the model is updated through CER, which consists of both safety and alignment rewards, not only relying on alignment reward. In addition, see reward signal changes throughout fine-tuning in **Figure 7**. The reward converging stably is empirical evidence against instability in practice. Moreover, more fair and powerful evaluator can be used to replace CLIP model in our method, e.g., CLIP convnext_xxlarge or advanced VLM models to reduce the potential bias.
>
> ## **Response to Q1:**
>
> Erasing unsafe concept in supervised fine-tuning (SFT) setting is non-trivial and has several drawbacks, e.g., SafeGen.
> 1. It requires training dataset including various of different unsafe prompts.
> 2. It requires ‘ground truth’ of safe images, e.g., blurry images.
> 3. Lack of generalization, the performance is bounded to the diversity of prompts in the training dataset.
>
> In contrast, our ForceForget learns “when faced with NSFW concepts, how to generate safe alternatives that still satisfy other objectives” with only few of unsafe prompts and without providing ground truth safe image.
> Moreover, as the main evaluation for nudity removal is based on NudeNet, using the same classifier for both training and evaluation might not a valid way to evaluate the performance of an approach and is also unfair to other erasing methods.
>
> ## **Response to Q2:**
> Our approach has the potential to be extended to SDXL or DiT models, as both models utilize cross-attention mechanisms. However, SDXL's U-Net structure is larger and uses two text encoders. Safety adapter's feature dimensions would need to be adjusted. For DiT, our safety adapter injection mechanism would shift from modifying the cross-attention in the U-Net to interacting with the Transformer modules, which would require more modifications.
> We will certainly consider to integrate our methods on these models in our future research.
>
> ## **Response to Q3:**
> In I2I task, we conducted experiments with nude image input + text input: *“a photo of a naked
> person”*. The erasure knowledge learned during T2I fine-tuning still applies because the model encountered various levels of conflicts during RL fine-tuning such as (generated NSFW image + filtered safe text), (generated ambiguous image + filtered unsafe text), strong conflicts (generated safe image + filtered unsafe text). See equation (8) in paper, unified attention space consists of image content representation and projected text representation from safety adapter, where the model learns to resolve conflicts between image content and safety objectives through attention re-weighting in cross-attention.

---

> > ### Comment · Reviewer_1zae · 2025-11-25
> > **Follow-Up Concern Regarding Q1:**
> >
> > Thank you for your clarification on how ForceForget addresses unsafe concept erasure in the SFT setting, particularly in comparison to prior methods like SafeGen.
> >
> > However, I have a follow-up concern regarding the reward design used during RL fine-tuning.
> >
> > As described in the paper, your method optimizes a combined reward function defined as:
> >
> > $$
> > \text{CER} = r_{\text{safe}} + r_{\text{align}}
> > $$
> >
> > My concern is that, unlike a loss function where gradients can be directly propagated to different components of the network, a scalar reward composed by simple addition may not guarantee that both objectives (safety and alignment) are simultaneously optimized in a balanced manner.
> >
> > - Since $r_{safe}$ and $r_{align}$ could have different scales or learning dynamics, how do you ensure that one objective does not dominate the learning process?
> > - Are any normalization, weighting, or adaptive scaling techniques applied to balance their contributions?
> > - Did you observe any trade-offs between safety and prompt alignment in your experiments?

---

> > > ### Author Response · Authors · 2025-11-27
> > > **To Follow-up Q1**
> > >
> > > Thanks for reviewer’ clarification. This is a insightful question as one of the fundamental challenges about multi-objective reward in RL.
> > >
> > > We ensure each component of safe and alignment reward in the same range [0, 1] by scaling before addition. This keeps both rewards operate within comparable numerical ranges to prevent reward collapse toward a single objective.
> > >
> > > Moreover, the overall reward is constrained by clipping [1] to stabilize training. As in Appendix, A.1, we set clip range into [-1.5, 1.5]. Since we didn’t search the optimal setting for this hyperparameter (which is out-of-scope of this work), there still potential for improvement in parameter selection. We leave this for the future work.
> > >
> > > We observed reward changes (safe and alignment rewards) during fine-tuning as following (we have updated the submission to add a visualization of reward, see in **Fig. 10 in Appendix A.7**):
> > >
> > > 1.	Early Stage: safety reward tends to reach zero, where unsafe score still higher than safe score.
> > >
> > > 2.	Mid Stage: both safety and alignment improve simultaneously as model learns safe alternatives.
> > >
> > > 3.	Late Stage: safety reward gradually reaches the maximum with slight improvement of alignment reward.
> > >
> > > According to our observations, fine-tuning around 60~70 epochs has a good trade-off between erasing and utility.
> > >
> > > [1] John Schulman, Filip Wolski, Prafulla Dhariwal, Alec Radford, and Oleg Klimov. Proximal policy optimization algorithms. arXiv preprint arXiv:1707.06347, 2017.
> > >
> > > We also conducted the additional ablation study to show the **necessity of alignment reward of CER for nudity erasing**. We evaluated NRR on 931 I2P (sexual) prompts and 1000 prompts from COCO30K on each 10 epochs when fine-tuning model with only safety reward or alignment reward. As in Table below, fine-tuning model with only safety reward can effectively erase NSFW concept while model’s prompt-alignment capability being reduced. In contrast, model struggled to erase NSFW concept by fine-tuning only with alignment reward.
> > >
> > > ## **Fine-tuning with only safety reward**
> > > | Epoch | NRR | CLIP Score |
> > > |---|---|---|
> > > | 10 | 90.76 | 31.0531 |
> > > | 20 | 90.12 | 30.6400 |
> > > | 30 | 95.70 | 30.2597 |
> > > | 40 | 96.24 | 21.1603 |
> > > | 50 | 96.13 | 29.8424 |
> > >
> > > ## **Fine-tuning with only alignment reward**
> > > | Epoch | NRR | CLIP Score |
> > > |---|---|---|
> > > | 10 | 91.94 | 31.1493 |
> > > | 20 | 92.48 | 31.0484 |
> > > | 30 | 93.56 | 31.0154 |
> > > | 40 | 93.45 | 30.9427 |
> > > | 50 | 93.89 | 31.0090 |

---

### Official Review · Reviewer_FwRB · 2025-11-04

**Soundness:** 3
**Presentation:** 3
**Contribution:** 3
**Rating:** 4
**Confidence:** 2

**Summary:**

This work proposed a novel concept erasing framework called ForceForget that utilizes a carefully designed erasing reward and safety adapter. The experiments show that the proposed method achieves the highest nudity-removal rate and robustness against Ring-A-Bell, P4D, and MMA attacks, while retaining competitive FID and CLIP scores. This work also shows that the proposed method can extend to erase general concepts including artistic styles and objects to show the generalization.

**Strengths:**

1. The proposed Concept Erasing Reward elegantly integrates a safety reward (driven by an NSFW classifier) and an alignment reward (via CLIP–BLIP caption consistency).

2. The experiment section is comprehensive. Benchmarks cover T2I and I2I, explicit and implicit NSFW prompts, red-teaming attacks, and even artistic-style and object erasure—providing a wide empirical view.

3. The proposed safe cross attention change is novel and effective.

**Weaknesses:**

1. The effectiveness of the method appears to depend strongly on the relative weighting between the safety and alignment rewards, yet the paper does not include an ablation or sensitivity study on these hyperparameters. It remains unclear how robust the approach is to different reward weightings or whether the same configuration generalizes across concepts.

2. Reinforcement fine-tuning of diffusion models can be resource-intensive compared to closed-form or lightweight LoRA-based erasure methods. However, the paper does not report training time, GPU hours, or efficiency comparisons, making it difficult to assess the practicality of the proposed approach.

**Questions:**

1. Could the authors provide an ablation study or sensitivity analysis on the relative weighting between the safety reward and alignment reward in the Concept Erasing Reward (CER)? This would clarify how robust the method is to reward scaling and whether the same configuration generalizes across different unsafe concepts.

2. Could the authors report the training overhead (e.g., GPU hours, iterations, or wall-clock time) and compare it with other baseline methods to assess the practical efficiency of the reinforcement fine-tuning approach?

---

> ### Author Response · Authors · 2025-11-21
> **Author Response**
>
> We thanks reviewer’s thoughtful comments, especially for weight settings of reward. We provide detailed responses to your concerns below.
>
> ## **Response to W1 and Q1:**
>
> In the paper, we didn’t specify the individual weight for safety and alignment reward, i.e., **1** as weight for both rewards, which is fixed during fine-tuning. To show the impact of safety and alignment reward weighting, we fine-tuned SD v1.4 for nudity erasing for 60 epochs on each setting, other training configurations kept the same as in paper. We evaluated NRR on 931 I2P (sexual) prompts and 3000 prompts from COCO30K. Due to the constraints of our lab’s computational resources, we tested on several settings. See Table below, it seems that weights for safety and alignment do not greatly affect prompt-alignment but erasing ability. The balanced setting (1,1) allow model achieve better easing performance under the same epoch.
>
> | Safety&Alignment | NRR | CLIP |
> |---------|-----|------|
> | 1,1   | 97.96    | 30.786 |
> | 1,2   | 91.41    | 30.900 |
> | 2,1   | 81.95    | 30.997 |
>
> For generalization across different concepts, we encourage reviewer to see **Response to Q4 for Reviewer FDCS**.
>
> ## **Response to W2 and Q2:**
> Our method is motivated by its potential for robustness erasing compared to closed-form or lightweight methods (see performance of robustness against attacks in Table 1). The computational cost is a valid practical concern in RL. In term of training overhead, our method takes, e.g., 12 GPU hours for fine-tuning 60 epoch (around **12 minutes** per epoch) in nudity erasing task. As other methods didn’t report training time and some of them didn’t provide training scripts, it is not feasible to have a fair comparison.

---

> > ### Comment · Reviewer_FwRB · 2025-11-24
> >
> > Dear Authors:
> >
> > Thanks for your responses and extra experiment results. Most of my concerns have been addressed and I have updated my scores.

---

> > > ### Author Response · Authors · 2025-11-27
> > > **Author Response**
> > >
> > > Thanks for raising scores. We appreciate that we were able to resolve the reviewer's concerns.

---

### Author Response · Authors · 2025-12-03
**Summary of Discussion Period**

Dear Area Chair,

We have provided a short summary of discussion for your kind attention.
In our work, we compared the proposed **ForceForget** with other **10** SOTA erasing methods. For general comparison, see in following Table. NRR on nude image as input in I2I task and average NRR cross three red-teaming tools for measuring erasing robustness.

| Method  | Publication     |Train only on textual data |Trained Component | Easing in I2I | Robustness |
|-----------|------------|------|----------|----|-|
| SLD (Max) | CVPR2023   | Yes  | /        |N/A | 59.1|
| ESD       | ICCV2023   | Yes  | UNet     |12.4| 81.1|
| SA        | NeurIPS2023| Yes  | UNet     |N/A | 62.2|
| CA        | ICCV2023   | Yes  | UNet&CA  |N/A | 76.8|
| SafeGen   | CCS2024    | No   | UNet     |18.8| 90.6|
| RECE      | WACV2024   | Yes  | UNet&CA  |28.6| 84.9|
| MACE      | CVPR2024   | No   | UNet&CA  |8.4 | 90.3|
| DuMo      | AAAI2025   | Yes  | CA       |N/A | 98.0|
| SAFREE    | ICLR2025   | Yes  | /        |N/A | 65.0|
| Co-Erasing| ICML2025   | Yes  | UNet&CA  |9.75| 85.3|
| **ForceForget**  |     | Yes   | CA      |**96.4**| **99.8**|


**ForceForget** shows several advantages as following:
• High erasing effectiveness.
• High preservation of human-related contents.
• Strong robustness against adversarial attacks.
• Generalized erasing in I2I task.

We thank reviewers for their constructive comments and would like to acknowledge that
all of them gave us **score 3 (good)** in terms of **Soundness**, **Presentation** and **Contribution**.
Most of reviewers pointed out some common questions or concerns as following:

**1. Hypermeters of CER.**
We have provided the additional experiments to show the impact of different settings, e.g.,
weights for alpha and beta in safety reward and general weights for safety and alignment rewards.

**2. Choice of Number of Tokens in Safe Adapter.**
We have conducted experiment to show the impact of different number of tokens for erasing effectiveness and image quality.

**3. Erasing Different Unsafe Concepts.**
We evaluated erasing effectiveness on multiple unsafe concepts erasing using Q16 on full I2P.


Overall, we identify most of questions are about ForceForget's component configurations and hyperparameter settings.
We believe our responses during rebuttal have thoroughly addressed all reviewer concerns.


Moreover, we would like to note the **Reviewer FwRB** raised the score from **4** to **6**, although this change was not explicitly mentioned. We hope the Area Chair is aware of this situation.

Thanks for your time and effort!

Best,

Authors

---

### Meta-Review · Area_Chair_jdhh · 2026-01-07

**Summary:**

This paper proposes ForceForget, an RL-based framework for concept erasure in text-to-image diffusion models. The method introduces a Concept Erasing Reward (CER) that balances a safety term (from an NSFW classifier) and an alignment term (via CLIP-BLIP caption consistency), and employs a Safe Adapter that modifies only a subset of token projections in cross-attention layers to selectively suppress unsafe semantics. The paper demonstrates strong empirical results across multiple tasks, outperforming ~10 state-of-the-art baselines in terms of safety, fidelity, and generalization.

Reviewers generally agreed that the problem is important, the experimental evaluation is comprehensive, and the RL formulation is a novel direction in concept erasure. However, several fundamental concerns were raised:

- **(1) Heuristic design of the Safe Adapter:**
  The Safe Adapter modifies a fixed number of tokens (e.g., last 4) in the CLIP embedding sequence, but the paper does not justify why these specific positions are effective or whether the mechanism relies on implementation-specific artifacts (e.g., padding structure or positional encoding).
- **(2) Scalar reward combination lacks theoretical grounding:**
  The CER combines safety and alignment rewards via simple addition. Reviewers questioned whether this scalar reward can reliably balance potentially conflicting objectives, especially in prompts where safety and semantic fidelity are at odds.
- **(3) Computational overhead:**
  RL fine-tuning incurs significant training cost (~12 GPU hours reported), but the paper does not compare this against lightweight alternatives.
- **(4) Limited multi-concept analysis:**
  While evaluated on multiple unsafe categories, the RL training signal is derived solely from nudity-related prompts.

**Reviewer Concerns:**

Following the rebuttal, several empirical questions, such as reward weighting, token count sensitivity, and training overhead, were adequately addressed. The authors also demonstrated consistent performance when swapping NSFW classifiers and provided ablation on multi-concept erasure using the Q16 evaluator.

However, two fundamental issues remain unresolved:

**First, the Safe Adapter’s token selection mechanism lacks principled justification.**
The choice to fine-tune only the last k tokens of the CLIP sequence is empirically tuned but not conceptually grounded. In Stable Diffusion, these terminal tokens typically correspond to [EOS] padding and carry minimal semantic content. It is therefore unclear how modifying them selectively suppresses unsafe semantics without degrading generation quality. This raises concerns that the method may exploit low-level architectural properties (e.g., attention masking patterns or positional biases) rather than achieving semantic-level intervention. Without analysis of which features are altered and how they relate to unsafe concepts, the approach remains a black-box technique with uncertain transferability.

**Second, the scalar combination of safety and alignment rewards offers no guarantee of balanced optimization under objective conflict.**
While the authors normalize both rewards to [0,1] and test a few weight settings, this does not address the fundamental limitation of scalarization in multi-objective RL: when safety and alignment pull in opposite directions, the policy may converge to a compromised state that is neither fully safe nor fully aligned. The paper provides no analysis of such failure modes or theoretical conditions under which the combined reward yields Pareto-optimal behavior. In safety-critical domains, such opacity in trade-off handling undermines trust.

These limitations undermine the method’s interpretability and trustworthiness, despite its strong empirical performance.

**Reviewer Scores:**

Reviewer FwRB (initially 4) raised their assessment to 6 after rebuttal, and Reviewer 1zae maintained a positive score (6). Reviewer FDCS (rating 4) expressed lingering concerns about adapter design and generalization that were partially alleviated. Given the absence of post-rebuttal engagement under ICLR 2026 guidelines, it is likely that Reviewer FDCS would retain a marginal stance.  The overall sentiment is borderline.

---

### Decision · Program_Chairs · 2026-01-26

Reject